# Knowledge, Attitude, and Practice Towards Antibiotic Use and Resistance Among Non-Medical University Students, Riyadh, Saudi Arabia

**DOI:** 10.3390/ijerph22060868

**Published:** 2025-05-31

**Authors:** Amen Bawazir, Abdullah Bohairi, Omar Badughaysh, Abdulmohsen Alhussain, Mohannad Abuobaid, Majd Abuobaid, Azzam Al Jabber, Yaman Mardini, Abdulaziz Alothman, Faris Alsomih, Abdullah AlMuzaini, Mohammed BaHamdan

**Affiliations:** Community Medicine Unit, Basic Medical Science Department, College of Medicine, AlMaarefa University, Diriyah 13713, Saudi Arabia; 211120387@student.um.edu.sa (A.B.); 211120330@student.um.edu.sa (O.B.); 202120441@student.um.edu.sa (A.A.); 211120066@student.um.edu.sa (M.A.); 211120065@student.um.edu.sa (M.A.); 171120029@student.um.edu.sa (A.A.J.); 211120269@student.um.edu.sa (Y.M.); 201120149@student.um.edu.sa (A.A.); 211120116@student.um.edu.sa (F.A.); 201120108@student.um.edu.sa (A.A.); 192120568@student.um.edu.sa (M.B.)

**Keywords:** antibiotic resistance, antibiotic misuse, public health education, health literacy, multidrug-resistant organisms (MDROs), antibiotic stewardship

## Abstract

Background: Antibiotic resistance (AR) is a global public health crisis, largely driven by the misuse and overuse of antibiotics. This study aimed to assess the level of knowledge, attitude, and practice (KAP) toward AR among non-medical university students in Riyadh, Saudi Arabia and identify factors that influence antibiotic use. Methods: A cross-sectional survey was conducted with 672 students from both public and private universities in Riyadh. Data were collected using a self-administered questionnaire covering sociodemographic characteristics and KAP regarding antibiotics. Results: The findings indicated that 59.1% of students had sufficient knowledge of antibiotics, while 60% had positive attitudes, and 60.6% exhibited responsible practices. However, 40.9% of students demonstrated insufficient knowledge, and factors such as age, gender, and field of study significantly impacted KAP outcomes. Females had 65.8% higher odds of demonstrating a positive attitude toward AR prevention, compared to males, and 52% higher odds of adopting appropriate practices than males. The primary source of information on antibiotics was healthcare professionals, particularly doctors. Conclusions: These findings underscore the need for targeted educational interventions to enhance awareness and promote the responsible use of antibiotics among university students, helping to mitigate the threat of antibiotic resistance.

## 1. Introduction

Antibiotic resistance has emerged as a critical global health challenge, driven by the widespread misuse and overuse of antibiotics, leading to the emergence of multidrug-resistant organisms (MDROs), which significantly complicate treatment options [1,2]. Antimicrobial resistance (AMR) poses an urgent global health crisis, contributing to approximately 700,000 deaths worldwide annually. Without effective prevention strategies, this figure could rise to 10 million deaths per year by 2050 [3]. Projections for the period between 2025 and 2050 indicate a dire scenario, with models estimating 39.1 million direct deaths globally due to antimicrobial resistance, along with an additional 169 million fatalities indirectly associated with its impacts [4,5].

Saudi Arabia, a Global Health Security Agenda (GHSA) member since 2015, has been actively addressing AR through its National Action Plan (NAP), launched in 2017. This comprehensive strategy aligns with the World Health Organization’s (WHO) Global Action Plan and focuses on multi-sectoral collaboration, evidence-based interventions, and enhancing awareness of AMR among both the public and healthcare professionals [6,7]. The NAP is structured around five sub-committees dedicated to raising awareness, monitoring resistance patterns, strengthening infection control, optimizing antibiotic use, and advancing research.

In 2018, the Saudi Ministry of Health (MOH) implemented a national policy requiring a prescription for all antibiotic sales in pharmacies, aiming to curb the misuse of antimicrobials. Violations of this regulation carry severe penalties, including license termination, substantial fines, and potential imprisonment [8]. Additionally, Saudi Arabia has joined the WHO’s Global Surveillance Program, further reinforcing its commitment to combating AMR [9].

A national study on Gram-positive bacteria in Saudi Arabia revealed alarming resistance trends. Among Staphylococcus aureus isolates, 32% were methicillin-resistant (MRSA), while Streptococcus pneumoniae demonstrated penicillin G resistance (33%) and erythromycin resistance (26%) [10]. These findings highlight the pressing need for sustained antimicrobial stewardship efforts and strict enforcement of antibiotic regulations to mitigate the growing threat of AMR in the region.

Understanding public knowledge, attitudes, and practices regarding antibiotics is essential for developing effective strategies to combat AR [11]. Recent studies indicate that many individuals lack adequate knowledge about proper antibiotic use, contributing to the spread of resistance. Al-Shibani et al. (2017) found that self-medication with antibiotics, even for minor illnesses like sore throats, is common in Riyadh [12]. Additionally, many individuals discontinue antibiotics as soon as they feel better rather than completing the full course, further exacerbating AR risks. Al Nasser et al. (2021) reported that while people may possess some knowledge about antibiotics, they often exhibit negative attitudes toward their use [13]. For example, while many refrain from sharing leftover antibiotics—considered a positive behavior—negative attitudes toward antibiotic use persist. Similarly, Mostafa et al. (2021) found that Egyptian university students demonstrated low health literacy and limited knowledge about antibiotics, increasing the likelihood of misuse [14].

Including non-medical university students in this study is critical due to their heightened risk of antibiotic misuse, lack of formal medical training to guide appropriate use, and potential influence on public health behaviors. Younger demographics, particularly university students, are globally recognized as high-risk groups for AR. Studies from England, Italy, and Cyprus highlight this trend, demonstrating that younger individuals report poorer antibiotic knowledge and higher misuse rates [15,16,17]. These deficiencies in health literacy and antibiotic education underscore the urgent need for targeted educational interventions [18]. Despite growing research on antibiotic use in Saudi Arabia, particularly in clinical and community settings [17,19,20,21,22,23,24,25,26], university students—especially those from non-medical backgrounds—remain understudied, despite their critical role in shaping public health behaviors. Studies exploring self-medication, antibiotic sharing, and misconceptions among students are scarce, all of which contribute to antimicrobial resistance [27]. A literature review identified only three studies on antibiotic KAP among healthcare students [9,10,11], with no published research on non-medical university students in Saudi Arabia [28,29,30].

Addressing this research gap is crucial, as it hinders the development of targeted educational interventions and youth-focused campaigns that align with the Kingdom’s National Action Plan to Combat AR [31]. Further investigation into this demographic is essential for promoting responsible antibiotic stewardship and mitigating resistance on university campuses and beyond. As an integral and educated segment of society, students influence public health behaviors and community practices. However, due to limited antibiotic education, non-medical students frequently misuse antibiotics—taking them without prescriptions, using leftover medication, sharing antibiotics, or discontinuing treatment early. These behaviors increase the prevalence of resistant infections, leading to harder-to-treat diseases, higher healthcare costs, and increased outbreak risks on university campuses and within the broader community, posing a significant public health threat [27].

This study aimed to assess knowledge, attitudes, and practices toward AR among non-medical university students in Riyadh, Saudi Arabia, and to identify factors influencing antibiotic use. Focusing on non-medical students is critical, as they represent a high-risk demographic for antibiotic misuse due to their lack of formal medical training, potential role in shaping public health behaviors, and limited awareness of AR consequences. By evaluating these factors, the findings can inform targeted educational interventions to promote responsible antibiotic practices, align with national antimicrobial stewardship goals, and mitigate AR spread.

## 2. Materials and Methods

### 2.1. Ethical Consideration

This study adhered to strict ethical guidelines. Verbal informed consent was obtained from all participants, detailing the study’s purpose, procedures, risks, and benefits. Participants were assured of their voluntary participation and the right to withdraw at any point. Data privacy was protected through secure storage, limited access to authorized personnel, and anonymization via unique identifiers. The research received ethical approval from the IRB of AlMaarefa University (Ref. No: IRB23-040, date: 21 April 2023).

### 2.2. Study Design and Settings

A cross-sectional design was used to measure the level of awareness of antibiotic misuse that leads to AR among students in public and private universities of Riyadh, Saudi Arabia. Riyadh is the capital of Saudi Arabia, with a currently estimated population of 8.6 million, according to the General Authority of Statistics in 2019 [32].

#### Inclusion/Exclusion Criteria

All non-medical students in any public or private university in Riyadh, regardless of their nationalities, genders, or fields of study, were included, whereas visiting students or medical students were excluded from the study.

### 2.3. Sample Size

Raosoft sample size calculator was used to determine the minimum required sample size [33]. Based on the calculation with a 5% margin of error, a 95% confidence level and a 32% prevalence rate of awareness with AR was estimated, based on a previous study conducted among adults (aged 18–>65 years) who were aware of AR [12]. The effective sample size calculated was 334; however, a larger sample size of 501 students was eventually included in the study, yielding approximately 1.5 times the initial sample size projection (1.5 design effect).

#### 2.3.1. Sample Technique

A multistage sampling method was used, first stratifying Riyadh universities as public or private (three universities for each). The sample was then proportionally allocated to each university. A convenience sampling was employed to select participants based on accessibility and willingness to participate on campus, continuing until the required sample size was met. The sample also proportionally reflected gender distribution according to official statistics, with female student ratios varying across institutions.

#### 2.3.2. Data Collection Instrument

A structured, paper-based, self-administered questionnaire—designed based on validated instruments from prior studies [16,31,34,35,36,37,38]—was distributed as printed copies to target participants across multiple university campuses. This in-person approach ensured direct participant engagement, comprehensive responses, and efficient data collection. Data collection was conducted from 4 September to 19 October 2023 by a team of trained students from the College of Medicine at AlMaarefa University, who followed standardized protocols for accuracy and consistency. It was piloted among 5% of the sample (35 students). The questionnaire was then reviewed and assessed by 3 subject experts for its content, design, relevance, readability, and comprehension. A content validity ration (CVR) was calculated for each domain, and all domains reported 0.73 for knowledge domain, 0.68 for attitude domain, and 0.76 for practice domain for Cronbach alpha coefficient score. Domains with less than 0.07 were reviewed carefully, and targeted questions were modified or deleted accordingly. Three questions related to knowledge from the previous literature were found not appropriate to the local context and then not included in the final version of the questionnaire.

The final version of the questionnaire comprised 53 questions, structured into five main sections: sociodemographic characteristics (5 questions), knowledge of antibiotics (13 questions), attitudes toward antibiotic use and the severity of antibiotic misuse (12 questions), practices related to antibiotic use (18 questions), and sources of antibiotic knowledge and information (5 questions). Knowledge-related questions were scored with a binary system, where incorrect or uncertain (“don’t know”) responses received a score of 0, while correct answers were awarded 1 point. The attitude and practice sections utilized a five-point Likert scale (strongly disagree to strongly agree), with responses scored from 1 (least appropriate) to 5 (most appropriate). To account for unfavorable statements, scores for such items were inverted during analysis. The attitude section had a possible score range of 12–60, while the practice section ranged from 18–90. These raw scores were standardized to a 0–100 scale (0 = worst, 100 = best) to enable uniform interpretation across domains. Criteria for correct/incorrect answers were determined through prior literature and expert validation during questionnaire development. This scoring framework ensured consistency in evaluating participants’ antibiotic-related attitudes and practices, aligning with established methodologies to enhance reliability and comparability of results.

Knowledge of AR was assessed using thirteen statements covering key aspects of antibiotic use, such as their effectiveness in treating bacterial versus viral infections, the potential transmission of resistant bacteria between individuals, and their perceived efficacy for common illnesses. Additional statements addressed misconceptions, including the use of antibiotics for viral conditions (e.g., colds, flu), pain relief, fever reduction, and availability as over-the-counter drugs. The questionnaire also included items on awareness of antibiotic side effects, the consequences of overuse leading to resistance, and the misconception that resistance is trivial or solely related to allergies. These statements aimed to assess both clinical understanding and societal beliefs about antibiotics, emphasizing the public health risks associated with misuse.

Attitudes toward antibiotic use were evaluated through twelve questions, including statements such as trusting a physician’s decision when choosing not to prescribe antibiotics and avoiding unnecessary antibiotic use. Some questions reflected tendencies toward misuse, such as believing antibiotics should be accessible without a prescription, sharing them with family members without medical consultation, or using them to prevent the worsening of illness. Other statements examined self-medication behaviors, such as taking antibiotics based on past similar symptoms (e.g., toothache, gastrointestinal issues), using expired antibiotics, or discontinuing antibiotic courses prematurely upon feeling better. Additional questions assessed understanding of antibiotic side effects and the perceived efficacy of alternative medicine as a substitute.

Practices related to antibiotic use were assessed through questions identifying negative behaviors, such as non-prescription use, sharing antibiotics, switching doctors to obtain non-prescription antibiotics, using antibiotics for symptoms like coughs without medical consultation, and relying on advice from non-medical sources. Positive practices were evaluated through questions about using antibiotics only as prescribed, following pharmacists’ recommendations, seeking medical consultation, reading medication labels, completing prescribed courses, checking expiration dates, and avoiding antibiotics obtained from relatives without prescriptions.

The study converted all antibiotic-related knowledge, attitude, and practice scores into percentages to standardize scoring and classify proficiency levels. A 60% proficiency threshold—determined through pilot study analyses (mean and median of knowledge scores)—was adopted to assess adequacy across domains. This benchmark aligned with Saudi Arabia’s academic passing standard (60/100) and mirrored methodologies in prior research [34,39], ensuring methodological consistency and contextual relevance. The 60% cutoff provided an objective classification of “sufficient” (scores > 60%) or “insufficient” (scores ≤ 60%) outcomes, enhancing interpretability while reflecting both local educational norms and established study frameworks. Participants scoring above 60% were deemed to have adequate KAP, whereas those at or below this threshold were categorized as having insufficient understanding or behaviors in antibiotic use.

#### 2.3.3. Quality Assurance

The questionnaire was initially written in English; then, with the assistance of a language expert, it was translated into the regional tongue (Arabic) and then back into English to maintain consistency. Although the dataset contained minimal missing values, specifically about ten instances within the attitude assessment, mean imputation was employed to address these gaps. This approach ensured the integrity and completeness of the analysis.

#### 2.3.4. Data Presentation and Analysis

All the questionnaires were reviewed before entering the data into the analysis program. Statistical analysis was done using a statistical package for the social sciences (SPSS) v.26.0. (Armonk, NY, USA: IBM Corp.) to generate tables and charts. Results were expressed as percentages, means, and standard deviations (SD) accordingly. The Shapiro–Wilk test was used to assess normality assumptions, as it is reliable for small samples and can detect normality deviations precisely.

This study used Pearson’s Chi-Square test to explore the relationship between antibiotic knowledge, attitudes, and practices and demographic factors, after confirming the test’s assumptions, notably that expected cell frequencies were above five. A *p*-value of less than 0.05 was considered statistically significant for interpreting the findings.

## 3. Results

### 3.1. Demographic Characteristics:

As depicted in Table 1, among the 672 participants involved in this study, the mean age of the students was 20.2 years [±standard deviation (SD) 1.8]. Notably, the majority of participants were in the age group 19–20 years old (36.8%), unmarried (96.0%), female (60.0%), and of Saudi nationality (94.5%). In terms of university affiliation, IMSU had the highest representation, with 54.0% of participants. When it came to the field of study, the majority were enrolled in business programs (34.1%).

### 3.2. Knowledge, Attitude, and Practice of Antibiotic Resistance

The mean knowledge score related to AR was 63.97 ≈ 64 (±8 SD, range ≈ 49). Specifically, 43% (289 out of 672) of the participants demonstrated an adequate understanding of antibiotic resistance, while the remaining 57% (383 out of 672) exhibited an inadequate knowledge of this topic.

The findings in Table 2 indicate that the majority of university students demonstrated sufficient knowledge (59.1%), positive attitude (60.0%), and appropriate practice (60.6%) regarding antibiotic resistance. However, a notable minority (≈40%) lacked critical understanding or engagement, highlighting gaps in education and behavior. While overall awareness and behavior appear positive, nearly four in ten students lack proper understanding and engagement in AR practices.

Table 3 illustrates the analysis of the association between sociodemographic characteristics and sufficient knowledge, positive attitude, and appropriate practice of antibiotics among university students and reveals several significant patterns.

Age significantly impacted antibiotic-related attitudes and practices among students. Notably, students aged 19–20 demonstrated the highest rates of insufficient knowledge and unsafe behaviors compared to other age groups. This association was statistically supported, with significant *p*-values for attitudes (*p* = 0.003) and practices (*p* = 0.005).

Gender shows a significant association with attitudes (*p* = 0.005), with males exhibiting more insufficient attitudes than females, though no significant differences were found in knowledge (*p* = 0.343) or practices (*p* = 0.079). The field of study was significantly associated with all three aspects: knowledge (*p* = 0.003), attitude (*p* = 0.004), and practices (*p* < 0.001). Students from literary fields were the most likely to have insufficient knowledge, attitudes, and practices, while those in science and engineering showed better outcomes. However, nationality and university affiliation showed no significant associations across knowledge, attitude, or practice, with all *p*-values greater than 0.05.

Table 4 illustrates the analysis from multivariate backward regression test of factors associated with students’ sufficient knowledge, positive attitudes, and appropriate practices regarding AR, revealing several significant findings. Students aged 23–24 years have a 59% lower likelihood of having a positive attitude (OR = 0.410, 95% CI: 0.227–0.741, *p* = 0.003) and are significantly less likely (58.4%) to engage in appropriate antibiotic practices (OR = 0.416, 95% CI: 0.229–0.755, *p* = 0.004) to prevent AR compared to the youngest group (17–18 years). Females have 65.8% higher odds of demonstrating a positive attitude toward AR prevention compared to males (OR = 1.658, 95% CI = 1.170–2.350, *p* = 0.004) and 52% higher odds of adopting appropriate practices than males (OR = 1.520; 95% CI: 1.070–2.160; *p* = 0.019). Field of study also played a role. Business students and engineering students were significantly more likely to have a positive attitude compared to sciences students (OR = 1.988, 95% CI = 1.053–3.753, *p* = 0.034), (OR = 2.265, 95% CI = 1.127–4.552, *p* = 0.022), meaning they were more likely to have a better attitude towards antibiotic resistance. Moreover, engineering students were significantly more likely to follow proper antibiotic use practices than sciences students (OR = 2.243, 95% CI = 1.082–4.651, *p* = 0.030). On the other hand, literary students had 52.2% lower odds of sufficient knowledge compared to sciences students.

Other factors such as nationality and university affiliation were not significantly associated with knowledge, attitude, or practice differences. Overall, the study highlights the need for interventions to improve students’ understanding of AR and promote appropriate practices.

The primary source of information about antibiotic use among participants was their doctor, with 76.8% (516 participants) relying on this source (Figure 1). Pharmacists were the second-most common source, consulted by 9.4% (63 participants). Family/siblings or friends provided information for 7.0% (47 participants), while social media was a source for 6.8% (46 participants).

## 4. Discussion

This study aims to assess the level of awareness about AR. Findings from this study demonstrated that the respondents’ general understanding of antibiotics and AR was 59.1%, which is consistent with the existing literature from United Arab Emirates (59%) and Malaysia (61.1%) among non-university students [22,27,35] but still reveals critical gaps. This is in contrast to other studies from Jordan, Thailand, Zambia, Brunei, Sudan, and Cyprus, where non-medical students exhibited lower levels of knowledge regarding antibiotics and antimicrobial resistance [20,21,23,24,25,26,28,29]. However, other studies from Italy and Nepal reported a very high level of knowledge to antibiotic use compared to our findings (84.8%, 94%) [30,31].

In our study, factors like age, gender, and field of study influenced these outcomes, which revealed that older students (aged 23–24) had better attitudes and practices, since they went deeper into medically related educational materials, which is consistent with Wei Tiong (2020) [27], while female students were more likely to have sufficient attitudes and practices compared to males. Students in literary colleges displayed notably reduced antibiotic knowledge compared to their science counterparts (OR = 0.478; 95% CI = 0.252–0.908). This disparity aligns with the observation that science and engineering students, benefiting from biology/microbiology-rich curricula and practical lab work, demonstrate enhanced antibiotic understanding and practices, thus contributing to improved public health literacy.

Unlike a prior study where women had significantly lower odds (OR = 0.57, 95% CI = 0.465–0.699, *p* < 0.001) compared to men [40], these results differ. Such contradiction is probably due to the inclusion in their sample with all age groups rather than a young population of university students [41].

Our study revealed a lack of knowledge regarding the safe use of antibiotics and inadequate medical practices related to their consumption among the population. It is widely recognized that the uncontrolled use of antibiotics can have serious and significant adverse effects [42,43], including the emergence and prevalence of resistant microbial strains, which could be attributed to a lack of health education campaigns and limited exposure to healthcare educators. Initiating a health awareness campaign can assist legislators in making informed decisions to increase awareness.

Since 2018, Saudi Arabia’s Ministry of Health has enforced regulations under the Health Practice Law, prohibiting pharmacists from dispensing antibiotics without a valid prescription from a licensed physician. The policy has contributed to a decline in antimicrobial use within community pharmacies across the country [41]. However, despite a minimal reduction in non-prescription antimicrobial utilization following policy implementation, strengthening educational programs and increasing awareness among university students and the general public could further enhance the effectiveness of these measures [6].

The present study showed that university non-medical students’ practice toward antibiotic use was approximately 60.6%. This suggests that the majority of participants had a moderate level of practice towards antibiotic use, similar to the reported findings from university student in Nepal [24] and in line with the almost around the global average of 56%, according to the 2020 WHO survey [19]. Inappropriate antibiotic use, which is concerning, fuels antibiotic resistance. Age is one factor linked to less responsible antibiotic behaviors, suggesting certain groups are more prone to misuse.

This study found that non-medical students at Riyadh universities demonstrate strong awareness of antibiotic use information sources, with 76% relying primarily on physicians—a rate markedly higher than the 38% observed among Malaysian university students [27]. This reliance on physicians is encouraging, as they offer credible, evidence-based guidance. However, expanding awareness to include pharmacists—equally qualified to educate on antibiotic use and resistance—is critical. Students may lack awareness of pharmacists’ expertise, highlighting an opportunity to diversify accessible, trusted healthcare sources for public education. The Center for Disease Control and Prevention (CDC) and other global health agencies advocate for integrated public health strategies to fight MDROs, particularly among university students. Combining prevention and response interventions is essential for effective control in healthcare settings [44].

In our Riyadh-based sample, participants were predominantly Saudi nationals, reflecting the local university demographics. While our study primarily focused on university students within Riyadh, we acknowledge that nationality could indeed play a role. Factors such as cultural health beliefs, access to healthcare information, and variations in national health policies could all contribute to differences in awareness. Future research could explicitly explore nationality’s role by comparing diverse populations or incorporating multinational cohorts.

The study highlights persistent knowledge gaps about AR among non-medical university students in Riyadh, despite moderate general awareness. Attitudes and practices often deviated from guidelines, particularly among younger students (19–20 years), who exhibited significant knowledge deficiencies (*p* < 0.05), pointing to age-related educational disparities. Students in science and engineering disciplines demonstrated stronger knowledge and more responsible practices than peers in other fields, underscoring the influence of scientific training in curricula. Risky behaviors—such as self-medication and sharing antibiotics without prescriptions—were widespread, revealing a mismatch between partial awareness and actual practices [13,45,46,47,48,49]. While these trends align with regional studies, they contrast with European data [15,16,40,50,51], likely reflecting cultural or institutional differences in health education. Collectively, the findings emphasize the need for targeted educational campaigns to improve antibiotic stewardship, addressing misconceptions and promoting responsible use within this population.

The results advocate for tailored educational programs to enhance antibiotic awareness, particularly in demographics where age, gender, and academic discipline significantly influence outcomes. Public health campaigns should prioritize the 17–24 age group, as university students represent a pivotal demographic shaping health behaviors, yet are often excluded from antibiotic stewardship initiatives. Younger students (19–20 years) emerged as a critical subgroup requiring early intervention during their formative academic years. Proposed strategies include integrating AR education into university curricula, launching campus-wide campaigns (e.g., digital outreach, mandatory health literacy modules), and partnering with pharmacies to reinforce responsible practices.

For engineering students, who demonstrated higher baseline knowledge, introducing an elective course on antibiotic awareness could leverage their scientific literacy and problem-solving skills. This course could cover resistance mechanisms, global antimicrobial resistance (AMR) trends, and ethical considerations in engineering contexts (e.g., pharmaceutical waste management). Collaborations with medical or public health faculties could foster interdisciplinary solutions, empowering engineers to design innovations (e.g., wastewater treatment systems) that mitigate AMR risks. Such initiatives would bridge technical expertise with public health priorities, positioning engineers as advocates for antibiotic stewardship.

### 4.1. Strength

Key strengths include a robust, gender-balanced sample of non-medical Riyadh students, enhancing statistical power and representativeness. A bilingual (Arabic/English) questionnaire ensured inclusivity and reduced language barriers—a novel regional approach. The focus on non-medical students addressed literature gaps, while standardized, WHO-aligned questions minimized variability and improved global comparability. These elements collectively strengthen insights into antibiotic awareness in an understudied demographic.

### 4.2. Limitation

This research has several limitations. First, the reliance on self-reported data introduces risks of recall bias (e.g., inaccuracies in recalling past antibiotic use) and social desirability bias (over reporting adherence to guidelines), potentially compromising the validity of responses. Second, the cross-sectional design precludes establishing causal relationships or analyzing temporal trends between awareness and behavioral factors. Third, the geographical restriction to Riyadh limits the generalizability of findings to rural or non-metropolitan populations in Saudi Arabia. Additionally, the absence of clinical validation—such as prescription audits or medical records—may lead to overestimations of appropriate antibiotic practices

## 5. Conclusions

This study reveals a significant lack of knowledge and understanding about AR among non-medical university students. However, some research questions should be included to investigate the specific aspects of AR that students are unaware of, examine how factors like age, gender, and field of study influence practices, evaluate the effectiveness of different educational interventions, and explore policy implications for addressing this issue. Targeted education is crucial to improve antibiotic awareness, especially among specific demographics. Public campaigns should involve pharmacists alongside physicians. Policymakers must implement multifaceted interventions addressing healthcare and patient factors. This study provides vital data for public health initiatives and informs effective strategies to promote responsible antibiotic use.

## Figures and Tables

**Figure 1 ijerph-22-00868-f001:**
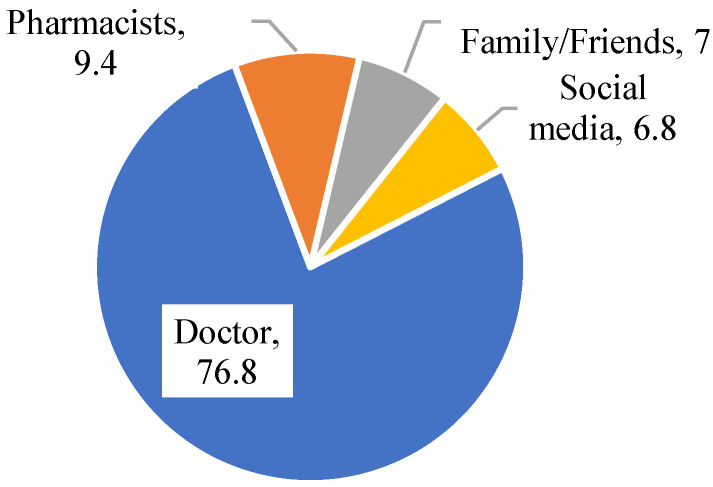
Sources of information about the participants’ antibiotic use.

**Table 1 ijerph-22-00868-t001:** Sociodemographic characteristics of the participants.

Variable	Categories	No.	%
Age groups (years)	17–18	140	20.8
	19–20	247	36.8
	21–22	202	30.1
	23–24	83	12.4
Gender	Male	269	40.0
	Female	403	60.0
Nationality	Saudi	635	94.5
	Non-Saudi	37	5.5
University	IMSU	363	54.0
	KSU	181	26.9
	PNAU	84	12.5
	PSU	11	1.6
	AFU	21	3.1
	AYU	12	1.8
Field of study	Business	229	34.1
	Literary	206	30.7
	Engineering	93	13.8
	Computer science	81	12.1
	Sciences	63	9.4
Marital status	Married	27	4.0
	Non-married	645	96.0

IMSU = Imam Mohammed Ibn Saud Islamic University; KSU = King Saud University; PNAU = Princes Noura bint Abdulrahman University; PSU = Prince Sultan University; AFU = AL Faisal University; AYU = Al Yamamah University.

**Table 2 ijerph-22-00868-t002:** Level of knowledge, attitude, and practice toward AR among the university students.

Total Knowledge	No.	%
Insufficient knowledge	275	40.9
Sufficient knowledge	397	59.1
Negative attitude	269	40.0
Positive attitude	403	60.0
Inappropriate practice	265	39.4
Appropriate practice	407	60.6

**Table 3 ijerph-22-00868-t003:** Association between sociodemographic characteristics and sufficient knowledge, attitude, and practice of antibiotics among university students.

Variable	Categories	Sufficient Knowledge			Sufficient Attitude			Sufficient Practices		
		No.	%	*p* Value	No.	%	*p* Value	No.	%	*p* Value
Age	17–18 years	88	62.9	0.109	92	65.7	0.003	94	67.1	0.005
	19–20 years	148	59.9		147	59.5		146	59.1	
	21–22 years	122	60.4		129	63.9		130	64.4	
	23–24 years	39	47.0		35	42.2		37	44.6	
Gender	Male	153	56.9	0.343	144	53.5	0.005	152	56.5	0.079
	Female	244	60.5		259	64.3		255	63.3	
Nationality	Saudi	372	58.6	0.280	380	59.8	0.780	384	60.5	0.838
	Non-Saudi	25	67.6		23	62.2		23	62.2	
University	IMSU	200	55.1	0.300	205	56.5	0.220	209	57.6	0.150
	KSU	118	65.2		120	66.3		121	66.9	
	PNAU	50	59.5		54	64.3		51	60.7	
	PSU	7	63.6		5	45.5		4	36.4	
	AFU	14	66.7		13	61.9		15	71.4	
	AYU	8	66.7		6	50.0		7	58.3	
Field of study	Business	136	59.4	0.003	151	65.9	0.004	147	64.2	<0.001
	Literary	101	49.0		106	51.5		99	48.1	
	Engineering	65	69.9		66	71.0		71	76.3	
	Computer sc	52	64.2		45	55.6		50	61.7	
	Sciences	43	68.3		35	55.6		40	63.5	

**Table 4 ijerph-22-00868-t004:** Factors associated with the students’ sufficient knowledge, positive attitudes, and appropriate practices regarding antibiotic resistance.

Variable	Categories	Knowledge			Attitude			Practices		
		OR	95% C. I	*p*-Value	OR	95% C. I	*p*-Value	OR	95% C. I	*p*-Value
Age	17–18 years	R	-	-	R	-	-	R	-	-
	19–20 years	0.992	0.633–1.554	0.971	0.792	0.501–1.251	0.317	0.740	0.466–1.175	0.201
	21–22 years	0.962	0.588–1.573	0.877	0.855	0.517– 1.414	0.543	0.819	0.493–1.361	0.441
	23–24 years	0.590	0.331–1.054	0.075	0.410	0.227–0.741	0.003	.416	0.229–0.755	0.004
Gender	Male	R	-	-	R	-	-	R	-	-
	Female	1.281	0.909–1.806	0.157	1.658	1.170–2.350	0.004	1.520	1.070–2.160	0.019
Nationality	Saudi	0.772	0.346–1.725	0.528	0.651	0.291–1.456	0.296	0.975	0.437–2.172	0.950
	Non-Saudi	R	-	-	R	-	-	R	-	-
University	IMSU	0.957	0.258–3.549	0.947	1.839	0.508–6.662	0.354	1.550	0.422–5.700	0.509
	KSU	1.132	0.304–4.209	0.853	2.722	0.751–9.872	0.128	1.741	0.473–6.407	0.404
	PNAU	0.770	0.197–3.020	0.708	1.751	0.457–6.704	0.413	0.993	0.255–3.870	0.992
	PSU	0.953	0.159–5.711	0.958	.801	0.140–4.595	0.803	0.337	0.056–2.032	0.235
	AFU	0.845	0.180–3.962	0.831	1.462	0.324–6.596	0.621	1.588	0.334–7.553	0.561
	AYU	R	-	-	R	-	-	R	-	-
Field of study	Business	0.749	.393–1.428	0.380	1.988	1.053–3.753	0.034	1.124	0.591–2.134	0.722
	Literary	0.478	0.252–0.908	0.024	0.996	.533–1.862	0.990	0.543	0.287–1.025	0.059
	Engineering	1.166	0.574–2.369	0.672	2.265	1.127–4.552	0.022	2.243	1.082–4.651	0.030
	Computer sc	0.886	0.435–1.804	0.738	1.158	0.582–2.304	0.677	1.053	0.521–2.126	0.886
	Sciences	R	-	-	R	-	-	R	-	-

R = reference.

## Data Availability

The datasets generated and/or analyzed during the current study are available from the corresponding author upon reasonable request.

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
