# Peer review of "Knowledge, Attitude, and Practice Towards Antibiotic Use and Resistance Among Non-Medical University Students, Riyadh, Saudi Arabia"

_ijerph, 2025, doi:10.3390/ijerph22060868_

Round 1
Reviewer 1 Report
Comments and Suggestions for Authors
I am glad to review this report. The authors addressed a relevant and timely public health issue: antibiotic resistance awareness among non-medical university students. I find the topic of this study is of potential interest to the international scientific community as the threat of antibiotic resistance is alarmingly increasing and awareness is one of our best tools to stop it. However, this reviewer thinks that the manuscript has a few pitfalls that must be overcome before acceptance.
Introduction:
While the introduction highlights the importance of antibiotic resistance (AR), it does not clearly articulate why non-medical university students are crucial demographic for this research. The introduction lacks strong rationale for the study. Several sentences reiterate the same point about AR being a global crisis without adding new insights. There is little discussion on the role of education in shaping antibiotic knowledge, attitudes, and practices (KAP). The authors ought to emphasise how the issue of inadequate awareness of AR has spread globally to put their research into a broader context. I suggest citing and commenting the following papers: doi: 10.1136/bmjopen-2021-055464, doi:10.1093/emph/eoaa028, doi: 10.3390/antibiotics12050897, doi: 10.3389/fphar.2022.903503, doi: 10.1038/s41598-020-60444-1.
Line 38: I don’t think that people can “self-prescribe” antibiotics. I suppose, “self-medicate” is more appropriate.
Materials and methods
2.2.2. Data Collection Instrument: How did you decide what variables to include in your questionnaire? Did you review pre-existing literature? Was the questionnaire pre-tested? If the questionnaire was pre-tested, do you identify possible questions that do not make sense to the participants or potential problem areas and/or deficiencies in the research instrument prior to implementation during the full study?
Line 89: the authors stated that a total of 44 questions were in the survey. Please, check. Summing up the questions from just the second, third, fourth, and fifth parts, there are 44 questions in total.
Line 92: Is “5 points Likert score as.” a typo? Please, check.
Line 92-94: I suggest, after saying that the survey instrument is divided into “the following 5 parts”, to list them one after one (not just the first part).
Lines 101-106: Please clarify on what basis a cut-off of 60% was chosen. Please provide further information about how you calculated the knowledge, attitude and practice scores. For example, what were the maximum and minimum scores? The possible answers were on a 5-point Likert scale? Please, clarify.
In lines 144-145, you stated that you “categorized as either "strongly agree" or "strongly disagree" depending on the statement.”. I suggest to also consider the option “agree” or “disagree” correct. Alternatively, clarify why you decide to score only "strongly agree" or "strongly disagree" as correct options.
Lines 122-127: I suggest adding that the study protocol received ethical approval in the main text.
Results
Line 148: Please, clarify the statement “A p-value of 0.05 was considered significant.”
Lines 149-157: Please, move the description of the knowledge questions in the methods section. In the results, describe the percentage of correct answers. I suggest doing the same for attitude (lines 158-168) and practice questions (lines 169-178).
Lines 179-185: The authors just repeat what have already stated in the material and methods section. I suggest deleting.
Lines 203-217 and Table 4: Please clarify if the outcomes are good or poor knowledge, positive or negative attitudes, and correct or wrong practices regarding AR. If necessary, revise the text if the outcomes are the ones stated in the title of Table 4.
Discussion
The discussion mostly restates the results without deeper interpretation or theoretical context. I suggest enhancing the discussion with a more critical analysis of findings, considering potential biases and contextual factors.
I believe the authors should compare the results with those of studies conducted in different contexts. Please, consider citing and commenting on the following papers: doi: 10.3390/antibiotics10091091, doi: 10.1371/journal.pone.0204878, doi: 10.1089/mdr.2018.0010, doi: 10.3390/ijerph18083930, doi: 10.2147/DHPS.S253301.
I believe that the discussion should delve more into practice, discussing the negative behaviors adopted by the sample to propose specific public health interventions. I suggest providing concrete policy recommendations to ensure the study’s implications extend beyond the academic field.
The role of receiving information from the physician or other healthcare workers (HCWs) deserves an in-depth discussion. Indeed, several studies support the educational role of HCWs or other medical figures.
Limits
The study’s limitations, such as self-reported bias and potential recall bias, are not adequately discussed.
Conclusions
While the conclusion mentions public education, there is little elaboration on how findings can inform interventions.
I suggest attaching the questionnaire as supplementary material.
Comments on the Quality of English LanguageI kindly suggest you have a fluent, preferably native, English-language speaker thoroughly copyedit your manuscript for language usage, spelling, and grammar.
Author Response
"Please see the attachment."

Reviewer 2 Report
Comments and Suggestions for Authors
I appreciate the opportunity to review the manuscript titled "Tailoring Antibiotic Resistance Education: Addressing Knowledge Gaps in University Students." The study addresses an important and timely public health issue—antibiotic resistance (AR) awareness among non-medical university students in Riyadh. The manuscript presents a structured study; however, I have several suggestions for improving clarity, methodology, and overall impact.
Title:
- The current title is too broad and could be more specific. Consider adjusting it to reflect the study’s focus on Knowledge, Attitudes, and Practices (KAP) related to antibiotic resistance among non-medical university students in Riyadh.
Abstract:
- The Results section would benefit from mentioning significant numerical findings to enhance clarity and impact. Clearly state key statistical outcomes, including percentages, means, standard deviations (SD), odds ratios (OR), confidence intervals (CI), and p-values, where relevant.
Introduction:
- The introduction is too brief and lacks global and regional context. To strengthen it, consider:
- Adding global and Saudi Arabia-specific statistics on antibiotic resistance.
- Discussing existing interventions or solutions to combat AR.
- Explaining why non-medical students play a role in antibiotic resistance and how their behaviours impact public health.
- Addressing whether Saudi Arabia has a National Action Plan (NAP) to combat antibiotic resistance and how this study aligns with it.
- The literature review should be expanded with more recent studies from Saudi Arabia. Consider referencing:
- "Knowledge, Awareness, and Perceptions Towards Antibiotic Use, Resistance, and Antimicrobial Stewardship Among Final-Year Medical and Pharmacy Students in Saudi Arabia."
- Reword the aim of the study to ensure it is clear, specific, and aligns with the results. The aim stated in the Introduction, Abstract, and Discussion should be consistent to avoid discrepancies.
Methods:
- Use "Inclusion/Exclusion Criteria" instead of "Study Subjects" for clarity.
- The sentence “A total of 501 sample size was obtained” is unclear—this seems like a results statement rather than a methodology description.
- There is no mention of whether the questionnaire was adapted from previous studies or developed from scratch. I suggest adding a subheading titled "Questionnaire Development" within the Methods section to clearly explain: Whether the questionnaire was adapted from validated tools used in prior studies or created specifically for this study. If adapted, provide references to previous studies. If developed from scratch, explain how items were generated, and whether they were based on expert consensus or literature review. Also, mention whether the questionnaire underwent pilot testing or expert validation before full implementation.
- The number of questions per section is inconsistent—ensure all details match the actual distributed survey.
- The manuscript does not clearly specify the specific aspects assessed in each section (knowledge, attitude, and practices). To improve clarity, the Methods section should outline the types of questions used in the survey for each category. Providing a brief description or example questions will enhance transparency and improve the study’s reproducibility.
- The multistage sampling technique is not explained in detail. How were participants randomly recruited ?
- Sample Technique section describing university population distributions does not add significant value and could be removed for clarity.
- The statement “44 questions with each question measured with the 5-point Likert scale” is inconsistent—how were demographic questions measured if a Likert scale was used? Ensure consistency.
- The sentence “High scores of 5 for correct answers and a minimum of 1 for the poor-scored answers. 5 points Likert score as.” is incomplete and lacks clarity.
- The sentence “The last part one question with 5 options was constructed to receive answers on the sources of information regarding the antibiotics was related to the sources of information obtained to build the knowledge and attitude as well as personal practices with the use of antibiotics.” is overly complex and grammatically incorrect. Simplify for better readability.
- The cut-off point (60%) for sufficient knowledge/attitude/practice lacks justification. Was this threshold based on prior research, or was it arbitrarily chosen?
- Revise translation terminology—"Arab" should be changed to "Arabic" for clarity.
- Explain how Cronbach’s alpha was calculated: Was it computed separately for each section or for the entire questionnaire?
- Handling of Missing Data – No mention of how incomplete responses were handled in the analysis.
- Normality Testing – The manuscript states that the Shapiro-Wilk test was used to assess normality, but why was this test chosen over alternatives (e.g., Kolmogorov-Smirnov test)? A brief justification is needed.
- Chi-Square Test Assumptions – It should be stated whether assumptions (e.g., no expected frequency <5 per cell) were checked before applying Pearson’s chi-square test.
- The manuscript states that ethical approval was obtained, but the IBR name, reference number, and approval date are missing.
Results:
- The results need proofreading and editing to correct awkward phrasing and unclear meanings.
- Some sentences belong in the Methods section rather than the Results (e.g., questionnaire details of knowledge questions). for example: "To assess knowledge of antibiotic-resistant, 13 statements were developed related to the nature of antibiotics to kill bacteria, whether antibiotics are medications to treat viral infections..."
- Use "The mean age" instead of "the average age" and present the age distribution in a table instead of listing categories.
- The phrase “a significant portion of the participants belonged to IMSU, accounting for (54.0%) of the total” could be simplified to “IMSU had the highest representation, with 54.0% of participants.”
- The range value (≈ 49) is unclear in "The mean knowledge score related to antibiotic resistance was 63.97 ≈ 64 (±8 SD, range ≈ 49)."—provide a precise minimum and maximum score range instead or remove it.
- "A p-value of 0.05 was considered significant." lacks context.
- The sentence "Other answers were in favor of the misuse of antibiotics were answered..." is grammatically incorrect and needs rewriting.
- In Table 2, standardize capitalization and add a column for Mean ± SD to provide deeper statistical insight beyond percentages.
- The phrase "appears to play a role" is vague—statistical significance confirms associations, so reword for clarity.
- The caption for Figure 1 does not match the provided figure—revise to accurately represent the content.
Discussion:
- The discussion repeats results instead of critically analyzing them—focus on interpretation rather than restating numbers and Compare findings with local and global studies on antibiotic awareness and resistance. Avoid numerical data in the discussion where possible—refer to key trends instead.
- Shorten long sentence in the first paragraph for better readability.
- Remove unnecessary commas after percentages and references.
- “Looking into the nationalities of the participants" is too informal—reword for academic tone.
- Explain why science and engineering students performed better in knowledge and practices. Consider discussing curriculum differences or exposure to scientific reasoning.
- The phrase "Twice as less likely" is incorrect—use "half as likely" or "significantly less likely".
- Expand and structure the strengths and limitations to be more precise and academically structured. Discuss potential biases (e.g., recall bias, self-reporting bias).
- In Conclusion Remove new research questions from the conclusion—these should be in a separate "Future Research" section before conclusion
References:
- Include additional citations from WHO, CDC, or systematic reviews on antibiotic resistance education. Incorporate recent similar studies from Saudi Arabia, such as: "Knowledge, Awareness, and Perceptions Towards Antibiotic Use, Resistance, and Antimicrobial Stewardship Among Final-Year Medical and Pharmacy Students in Saudi Arabia."
Author Response
"Please see the attachment."

Reviewer 3 Report
Comments and Suggestions for Authors
The authors conducted a study to assess the knowledge, attitudes, practices, and use of antibiotics among non-medical students. However, the study contains several errors that need to be addressed. For instance, the title states 'Addressing Knowledge Gaps,' which is inaccurate. The study does not identify knowledge gaps but rather presents conflicting values related to 'knowledge' among students. Also, the term 'knowledge' is not accurately applied, as the authors incorporate perspectives, attitudes, and usage patterns into its calculation, which appears inappropriate. Moreover, the discussion lacks sufficient references to provide a solid foundation for explaining the results. The authors must implement major revisions to improve the document's clarity and accuracy. Specific comments and suggestions can be found in the attached PDF
Here is a list of documents that must be referenced in the paper:
https://doi.org/10.3390/antibiotics11020197
https://doi.org/10.2147/IDR.S289964
https://doi.org/10.1016/j.sapharm.2007.04.004
https://doi.org/10.3390/antibiotics9060349
https://doi.org/10.1371/journal.pone.0122476
https://doi.org/10.3390/antibiotics8030154

English should be revised. Some errors were detected
Author Response
"Please see the attachment."

Reviewer 4 Report
Comments and Suggestions for Authors
I appreciate the effort put into this study and commend the authors for addressing such a critical issue. Implementing the suggested revisions will further strengthen the impact and clarity of this important work.

Author Response
"Please see the attachment."

Reviewer 5 Report
Comments and Suggestions for Authors
Please see my comments.

Must be improved.
Author Response
Responses in doc
Thank you sincerely for your thoughtful evaluation of my work. I deeply appreciate your kind words and the time you dedicated to reviewing it. Your positive feedback is both encouraging and motivating, and I am grateful for your recognition of the effort invested in this project.
Thank you for your thorough review and valuable feedback. We appreciate your attention to detail and the insightful comments provided. We would like to inform you that the issues raised have been addressed in the previous version of the manuscript submitted to the journal. We believe these revisions have effectively addressed the concerns raised. If there are any additional suggestions or areas that require further clarification, we are more than willing to make the necessary adjustments.

Reviewer 6 Report
Comments and Suggestions for Authors
Dear colleagues!
- I don't see the statistical significance of the study. Among millions of residents (by the way, how many of them are in the age group studied?) - 672 people - does not say anything
- A self-administered questionnaire was used — how exactly? Was it a Google form or a paper printout?
- what are the questionnaire questions?
- non-medical students — why this category?
- Do you think that nationality somehow influences awareness of antibiotic resistance?
- What does your research show? You surveyed the age group of 17-24 years old and what do you suggest? Introduce an elective course on antibiotic awareness for engineering students?
Conclusion: the material presented in the manuscript is certainly interesting, but corresponds to the level of a scientific and practical conference and does not carry the scientific load corresponding to the level of the journal.
Author Response
"Please see the attachment."

Round 2
Reviewer 1 Report
Comments and Suggestions for Authors
The authors have responded in detail to all of my previous comments and made clarifications to areas that were not previously clear to me - I think the manuscript is strong and have no further comments/suggestions to add. I would recommend publication without further revisions.
Author Response
Thank you sincerely for your thoughtful evaluation of my work. I deeply appreciate your kind words and the time you dedicated to reviewing it. Your positive feedback is both encouraging and motivating, and I am grateful for your recognition of the effort invested in this project. Should there be further opportunities to refine or expand upon this work, I will gladly incorporate your insights.
Once again, thank you for your support and constructive perspective.
Reviewer 2 Report
Comments and Suggestions for Authors
Thank you to the authors for their efforts in revising the manuscript. I appreciate the improvements made, which have notably strengthened both the clarity and rigor of the work. However, there are still a few minor areas that require further attention:
- Title: Please consider removing the abbreviation “KAP” from the title. It would be clearer and more appropriate to write out the full terms without abbreviation.
- Use of Abbreviations Throughout the Manuscript: Once an abbreviation is introduced, use only the abbreviation thereafter. There are several instances where the full term is repeatedly written alongside the abbreviation (e.g., “knowledge, attitudes, and practices (KAP) toward antibiotic resistance (AR)” is used multiple times).
- Ethical Approval: Please add the data of ethical approval (if applicable).
- Data Collection Date Format: The format used for the data collection period is unclear. The sentence currently reads: “Data collection was conducted from September 4 to October 19, 2023,”
- Number of Questionnaire Items: There seems to be a discrepancy in the reported total number of questions. The manuscript states that the final questionnaire comprised 54 questions, but based on the breakdown provided, it appears the total is 53. Please double-check and correct this inconsistency.
Author Response
- Title: Please consider removing the abbreviation “KAP” from the title. It would be clearer and more appropriate to write out the full terms without abbreviation.
Thank you for your comment, and accordingly the abbreviation (KAP) was removed from the title
2. Thank you for your valuable suggestion. It has been taken into consideration, and the abbreviated term has been used in place of the full wording as recommended.
3. Ethical Approval: Please add the data of ethical approval
Thank you for you, and it was added as:
"The research received ethical approval from the IRB of AlMaarefa University (Ref. No: IRB23-040, date: April 21, 2023)."
4. Data Collection Date Format: The format used for the data collection period is unclear. The sentence currently reads, “Data collection was conducted from September 4 to October 19, 2023.”
This part was clarified as the following:
A structured, paper-based self-administered questionnaire—
5. Number of Questionnaire Items: There seems to be a discrepancy in the reported total number of questions. The manuscript states that the final questionnaire comprised 54 questions, but based on the breakdown provided, it appears the total is 53. Please double-check and correct this inconsistency.
Thank you for your comment.
Yes, it was found the total were 53 NOT 54 and was corrected
Reviewer 3 Report
Comments and Suggestions for Authors
The authors have replied to all posed comments
Author Response
Thank you sincerely for your thoughtful evaluation of my work. I deeply appreciate your kind words and the time you dedicated to reviewing it. Your positive feedback is both encouraging and motivating, and I am grateful for your recognition of the effort invested in this project. Should there be further opportunities to refine or expand upon this work, I will gladly incorporate your insights.
Reviewer 4 Report
Comments and Suggestions for Authors
Thank you very much for your revision. All comments were revised point by point.
Author Response
I sincerely thank you for your considerate evaluation of my work. I truly value the time and attention you devoted to the review, and I am grateful for your encouraging and supportive feedback. Your recognition of the effort behind this project is highly appreciated.
Reviewer 6 Report
Comments and Suggestions for Authors
Dear colleagues!
It seemed to me that paragraph 2.2.5 was out of place. Also, numbers without leading zeros look strange - please add them.
In everything else, good job. Without exaggeration, the topic is relevant and very important with serious delayed consequences that people all over the world have already begun to feel. I wish you success in your further work!
Sincerely,
reviewer
Author Response
1. Paragraph 2.2.5 was out of place. Also, numbers without leading zeros look strange
Thank you for your comment. Upon revising the draft text, we found that paragraph 2.2.5 was missing. We have now corrected this appropriately.
Thank you sincerely for your thoughtful evaluation of my work. I deeply appreciate your kind words and the time you dedicated to reviewing it. Your positive feedback is both encouraging and motivating, and I am grateful for your recognition of the effort invested in this project.